# Retinal Microcirculation Measurements in Response to Endurance Exercises Analysed by Adaptive Optics Retinal Camera

**DOI:** 10.3390/diagnostics14070710

**Published:** 2024-03-28

**Authors:** Maria Anna Żmijewska, Zbigniew M. Wawrzyniak, Maciej Janiszewski, Anna Zaleska-Żmijewska

**Affiliations:** 1Faculty of Medicine, Student Scientific Society “Eye”, Medical University of Warsaw, 02-097 Warsaw, Poland; maria.zmijewska@student.wum.edu.pl; 2Faculty of Electronics and Information Technology, Warsaw University of Technology, 00-065 Warsaw, Poland; 3Faculty of Medicine and Dentistry, Medical University of Warsaw, 02-097 Warsaw, Poland; 4Department of Ophthalmology, Public Ophthalmic Clinical Hospital (SPKSO), Medical University of Warsaw, 00-576 Warsaw, Poland; azaleska@wum.edu.pl

**Keywords:** adaptive optics, retinal microcirculation, autoregulation, physical exercise, WRL

## Abstract

This study aimed to precisely investigate the effects of intensive physical exercise on retinal microvascular regulation in healthy volunteers through adaptive optics retinal camera (AO) measurement. We included healthy volunteers (11 men and 14 women) aged 20.6 ± 0.9. The heart rate (HR) and systolic and diastolic blood pressures (SBP, DBP) were recorded before and after a submaximal physical exertion of continuously riding a training ergometer. The superior temporal retinal artery measurements were captured using the AO—rtx1^TM^ (Imagine Eyes, Orsay, France) without pupil dilation. We compared measures of vessel diameter (VD), lumen diameter (LD), two walls (Wall 1, 2), wall-to-lumen ratio (WLR), and wall cross-sectional analysis (WCSA) before and immediately after the cessation of exercise. Cardiovascular parameter results: After exercise, SBP, DBP, and HR changed significantly from 130.2 ± 13.2 to 159.7 ± 15.6 mm Hg, 81.2 ± 6.3 to 77.1 ± 8.2 mm Hg, and 80.8 ± 16.1 to 175.0 ± 6.2 bpm, respectively (*p* < 0.002). Retinal microcirculation analysis showed no significant decrease in LD, Wall 1 after exercise: from 96.0 ± 6.8 to 94.9 ± 6.7 (*p* = 0.258), from 11.0 ± 1.5 to 10.4 ± 1.5 (*p* = 0.107), respectively, and significant reduction in VD from 118.5 ± 8.3 to 115.9 ± 8.3 (*p* = 0.047), Wall 2 from 11.5 ± 1.0 to 10.7 ± 1.3 (*p* = 0.017), WLR from 0.234 ± 0.02 to 0.222 ± 0.010 (*p* = 0.046), WCSA from 3802.8 ± 577.6 to 3512.3 ± 535.3 (*p* = 0.016). The AO is a promising technique for investigating the effects of exercise on microcirculation, allowing for the tracking of changes throughout the observation. Intensive dynamic physical exertion increases blood pressure and heart rate and causes the vasoconstriction of small retinal arterioles due to the autoregulation mechanism.

## 1. Introduction

The main objective of most clinical trials is to estimate the effect of a functional impact, e.g., pathological, physiological, or exercise-induced, compared to a control condition for easily measurable parameters in the human system, such as vascular flow or the vision system. The retinal microcirculation is the only microvascular bed that may be directly visualised and is described as “a window to the heart and brain” [1].

New diagnostic methods of the retina were gradually introduced in the second half of the 20th century. They included colour fundus photography, fluorescein and indocyanine, scanning laser ophthalmoscopy, angiography, and optical coherence tomography (OCT) [2,3,4]. OCT enables deep analysis of the retinal structures, including the choroid, and is presently the gold standard for diagnosing diseases in the posterior segment of the eye [4].

The emerging non-contact technologies have transformed the ophthalmologic in vivo longitudinal examinations of the eye structure with high resolution. A novel approach to non-invasive measurement based on the objective method of adaptive optics (AO) generates cellular and microvascular cross-sectional retinal imaging. AO enables a retinal examination at the cellular level [2,3,5,6]. Dimensional measurements on the local arteries for functional diagnosis of ophthalmic and systemic diseases with enhanced accuracy of 3–4 μm compared to classical OCT of 10 μm can be achieved [5,6]. AO enhances the image quality, eliminating wavefront distortions during examination [2,3,6]. The concept used in AO technology was first employed in astronomic telescopes to compensate atmospheric turbulences using deformable mirrors [3,6]. Enhanced imaging devices based on AO technology can be applied to assess small-sized arteriole microcirculation in the retina precisely.

rtx1^TM^ (Imagine Eyes, Orsay, France) is an AO retinal camera using infrared illumination (wavelength 850 nm), having a resolution of 3.5 μm and a field of view 4 × 4 degrees [3]. The device permits the acquisition of images in any retinal region, and the technology software AOdetect^TM^ (for the photoreceptor analysis, version 3.4) and AOdetect^TM^artery (for the retinal microvasculature analysis, version 3.4) enable repeated measurements in the same spots [3]. Examination using the rtx1^TM^ includes several parameters: the length of the eyeball, refractive error, pupil size, and the transparency of the ocular media. It is possible to adjust the retinal region depth under evaluation, which enables visualisation of retinal small arterioles and veins, photoreceptors (cones and rods), intraretinal deposits, neurosensory retinal atrophy, lamina cribrosa in the optic disc area and retinal microexudations, and microaneurysms [3]. The resolution of the images of the retinal microstructures in AO is comparable to those seen on histologic examination. It is possible because of the correction for aberration arising from various refractive surfaces in the eye within the AO imaging system [3].

Before the introduction of the first AO microscope, different techniques had been used to analyse the calibre of retinal vessels. Advances in fundus photography and techniques of retinal image analysis, especially static retinal vessel analysis (SRVA), have enabled objective and accurate retinal vascular calibre measurement with high reproducibility for detecting signs of retinopathy [7]. Other technologies, like scanning laser Doppler flowmetry (SLDF) or dynamic retinal vessel analysis of retinal vascular diameter in response to flickering light, made it possible to examine dynamic and functional aspects of retinal microcirculation [7]. Also, some studies have described spectral domain–optical coherence tomography (SD–OCT) applications and scanning laser ophthalmoscopy (SLO) for assessing the calibre of retinal vessels and optical coherence tomography angiography (OCTA) for retinal peripapillary and macular blood flow [8,9]. Still, OCT cannot distinguish the lumen from the vascular walls. Only SLDF and AO can estimate the wall thickness-to-internal diameter ratio (wall/lumen ratio, WLR) of retinal arterioles, and they are used in the analysis of microvascular changes in cardiometabolic diseases, such as hypertension, diabetes, and obesity, and to predict the risk of cerebrovascular events [2,6,10,11]. AO provides a more accurate and precise estimation of the WLR of retinal arterioles compared to SLDF and allows for differentiation between functional narrowing (due to autoregulation) and structural remodelling of a vessel [6,11]. Multiple studies reported a strong relationship between retinal vessel calibre in hypertension and diabetes mellitus [2,6,7,10,11,12]. In these two primary systemic diseases, micro- and macrovascular changes occur in long-term observation, and narrowing of retinal arterioles is a commonly described sign of retinopathy [2,6,11,12,13,14,15,16,17,18,19]. Data from the ARIC Study and Beaver Dam Eye Study showed that nondiabetic individuals with a smaller arterio-venous ratio (AVR) had a 50–70% higher incident diabetes risk, independent of other cardiovascular risks [16,17,18,19]. Microvascular changes in retinal arterioles are a significant predictor of coronary heart disease and risk of acute myocardial infarction, particularly in women [20].

Regular physical activity is recommended in the 2020 ESC Guidelines as the prevention strategy against cardiovascular diseases [21]. It was shown that exercise and changes in lifestyle behaviour allowed a 75% reduction in mortality due to chronic cardiovascular diseases [21]. Generally, the body’s physiologic responses to aerobic and resistance exercise episodes occur in the cardiovascular, respiratory, musculoskeletal, endocrine, and immune systems. These responses induced in controlled physiological settings in the laboratory can be carefully observed and measured based on precisely regulated exercises or stress load. The exercise intensity, ventilatory conditions, and measurement time influence the response of retinal vessel diameters to exercise. In a systematic review, Streese L. et al. analysed 22 original articles exploring the influence of physical activity on retinal microcirculation in children and adults [22]. The primary technique used for retinal microvascular evaluation in these studies was the SVA, with the central retinal arteriolar (CRAE) and venular (CRVE) diameter equivalents and the arteriolar-to-venular diameter ratio (AVR) [22]. Higher physical activity levels were associated with narrower CRVE in children and adults. Wider CRAE and a higher AVR were found in adults performing regular aerobic activities [22]. Hanssen et al. have shown that ten weeks of intensive physical training improved the retinal arteriole’s diameter and increased AVR in obese and lean runners [23]. The redistribution of blood to muscles and skin during exercise may also reduce the blood flow in the internal carotid artery, influencing fundus vessel density [22].

Less research has examined the acute effect of submaximal or maximal endurance exercises on retinal vessel diameters. There are differences in the results of studies measuring the size of retinal vessels just after exercise cessation. Nussbaumer M. et al. showed an increase in retinal arteriolar and venular diameter 5 min after high-intensity exercise cessation in healthy senior and younger participants [24]. These findings are similar to the redistribution of previous studies of dilated retinal vessels at delayed time points after dynamic exercise [25]. On the other hand, the mechanism of autoregulation in the retina should keep a constant retinal blood flow during and after dynamic physical activity due to the vasoconstriction of arterioles in response to an increase in intraluminal pressure [26,27,28,29]. This regulatory mechanism is called a myogenic response, also known as the Bayliss effect, and has been described using different imaging techniques [28,29,30,31]. In the study by Harris et al., the retinal blood flow has been examined with fluorescein angiography [28]. They found that the normal retinal hemodynamic response with unchanged retinal blood flow after intensive dynamic exercise included vasoconstriction that normalised flow and overall retinal blood transit [28]. Iester et al. assessed the retinal blood flow with a Heidelberg retina flowmeter (HRF) after exercise consisting of stair climbing. Their results showed decreased retinal blood flow in all analysed areas of retinal arterioles but without significance. They concluded that the autoregulation mechanisms allowed for stable blood flow in the retina [29]. Rueddel et al. measured the retinal vessel constrictions and dilations to flicker provocation of the RVA device after 20 min of intensive exercise. They revealed that arterial constriction to flicker was significantly increased within the first 1.5 h after exercise cessation [30]. The RVA device was also used in another study to analyse the retinal arteriole response to increased blood pressure in healthy subjects in five age groups, from 20 to 69 years old. The decrease in arterioles’ diameter after exercise was observed in patients below 40, representing the correct myogenic autoregulation model [31]. Only a few studies have analysed retinal arterioles’ diameter changes, and the devices used in them are needed to track vessel parameters such as walls, lumen, and total diameter.

The study aimed to investigate acute changes in retinal arteriole diameters precisely using a retinal adaptive optics camera, rtx 1^TM^, in response to submaximal dynamic exercise in healthy young adults.

## 2. Materials and Methods

The study was performed between 3 December 2021 and 11 March 2022, in the Department of Ophthalmology in cooperation with the Department of Cardiology and Cardiac Rehabilitation at the Medical University of Warsaw. The study was conducted under the tenets of the Declaration of Helsinki. The Bioethical Committee of the Medical University of Warsaw approved the study protocol (approval number KB/87/2015). Informed consent was obtained from all study subjects after the presentation of the study protocol. We included healthy volunteers (11 men and 14 women, all Caucasian), aged 20.8 ± 0.8. Study participants were healthy volunteers who led an active lifestyle. None of them practised sports professionally in the past or currently.

Exclusion criteria were an active smoking status and cardiovascular and respiratory medical history or blood pressure of 140/90 mm Hg or higher after 10 min of seated rest, pregnancy, or any ocular disease. The body mass index (BMI) was estimated as weight in kilograms divided by the square of height in meters.

### 2.1. Study Protocol

The examination protocol was the same for all participants, and one specialist carried out the AO retinal camera rtx1 examinations. The cardiologist attended during the entire examination time. An electroretinogram was carried out to confirm the absence of heart abnormalities.

The axial length of both eyes was taken before starting the physical exertion protocol with IOL Master 700 (Carl Zeiss Meditec AG, Hennigsdorf, Germany).

AO fundus images were obtained using rtx1™ (Imagine Eyes, Orsay, France—version 3.4) (Figure 1a). Image acquisition in a single position lasted approximately 4 s, during which 40 individual images were acquired [3]. Images of the supratemporal retinal arterioles were obtained in both eyes at a distance of 0.5—1-disc diameters from the edge of the optic nerve head. The size of analysed arterioles was between 70 and 130 μm. AO examinations were performed without any drops (dilating or anaesthetic, which might influence the examination results). The following parameters were assessed to evaluate vessel morphology: lumen diameter (LD), two walls’ thickness (WT1, WT2), and vessel diameter (VD). VD resulted from the single arteriolar wall (WT) plus the vessel lumen (LD) and single arteriolar wall thickness (WT): VD = WT + (WT + LD). The wall-to-lumen ratio (WLR) was automatically calculated as WLR = 2 × WT/LD, and VD and LD usage evaluated the vascular wall’s cross-sectional area (WCSA). The WLR and WCSA were obtained automatically from the AO artery detection software (version 3.4) [3]. All the retinal parameters were measured with the best quality three times on the scan. The arithmetic mean of these three values was used in the statistical analysis. The same retinal arterioles were measured twice: first before the exercise and second immediately after, during the two minutes following the cessation, with 80% maximal heart rate (HR), depending on age.

Physical exertion consisted of continuously riding a training bike ergometer (ergoline, Ergoselect 600, Reynolds Medical) (Figure 1b), increasing the resistance level every 2 min. At each level, the pedalling speed had to be constant. All participants reached 80% of the maximum theoretical HR, defined by the following equation: (220 − Age). In this way, exercise was considered maximal in the cardiologic stress protocol and was performed under the supervision of the cardiologist. None of the included participants disrupted exercise because of intense discomfort. No incident was reported during the exercises.

Heart rate and systolic and diastolic blood pressure (SBP, DBP) were recorded at rest after 10 min of being seated and on the bike at 5, 10 min of exercise and within the first 2 min after cessation of the effort, by using a standard blood pressure cuff on the left arm at heart level. Oxygen saturation at rest was collected using a pulse oximetry sensor placed on the fingertip. All patients had an oxygen saturation that exceeded 98% before exercise, but unfortunately, it was impossible to obtain reliable measures during exercises.

The rtx1^TM^ retinal camera and ergometer were located in the same room, and data were collected as quickly as possible after the cessation of exertion.

### 2.2. Statistical Analysis

For all findings, the selected level of statistical significance was a two-tailed *p* < 0.05. Independent *t*-tests were used to assess the difference between all the measurement parameters. Descriptive statistics (mean ± standard deviation) were calculated for all measurement parameters. A regression generalised linear model (GLM) was used for multivariate analysis. At the same time, Spearman rank correlation and Pearson correlation coefficient (r) were employed in univariate analysis to examine the association between continuous variables (WLR, WCSA, blood pressure parameters, and BMI) depending on data normality. The coefficient of determination (R^2^) assesses model fit. Statistical analyses were generated with Statistica^TM^ v. 13.2, TIBCO Software Inc., Palo Alto, CA, USA, (2017) TIBCO (https://docs.tibco.com, accessed on 2 January 2024.).

## 3. Results

The mean BMI was 23.1 ± 4.3 (kg/m^2^). The axial length was 24.1 ± 0.8 (mm) for right eyes and 24.1 ± 0.9 (mm) for left eyes (*p* > 0.05). Because there were no statistical differences in axial lengths between the right and left eye, only measurements from the right eyes were considered in the statistical analysis. The mean time of the cycling episode to achieve 80% of the maximal HR was 10.1 ± 2.9 min.

The mean cardiovascular parameters at rest were HR 80.8 ± 16.1, SBP 130.2 ± 13.2, DBP 81.2 ± 6.3. HR and SBP increased significantly after exercises: 175.0 ± 6.2 (*p* < 0.001) and 159.7 ± 15.6 (*p* < 0.001), respectively, while DBP decreased to 77.1 ± 8.2 (*p* < 0.022) (see Table 1).

The mean LD was 96.0 ± 6.8, VD 118.5 ± 8.3, WLR 0.234 ± 0.02, and WCSA 3802.8 ± 577.6 in the supratemporal artery of the right eye at rest. The mean LD, Wall 1, did not change significantly after physical effort but slightly decreased. The VD, Wall 2, WLR, and WCSA decreased significantly after exercises (*p* < 0.05) (see Table 2).

The contraction of smooth muscles in the retinal arteriole’s wall, which leads to a reduction in its thickness, is involved in the myogenic mechanism of retinal autoregulation in response to changes in pressure. There was a significant decrease in vessel diameter but not a significant lowering of lumen diameter. Thickening of arteriole walls is found in vessel remodelling due to vascular and metabolic diseases but not in physiological mechanisms.

Univariate regression analysis showed that BMI was significantly associated with WLR measured before exercise (0.423, R^2^ = 0.183) and the systolic blood pressure (SBP) change after exercise (−0.405, R^2^ = 0.190). Diastolic blood pressure (DBP) change was significantly associated with the shift in WCSA after exercise (−0.441, R^2^ = 0.194).

Multivariate regression analysis confirmed the independent association of BMI with WLR before exercise (−0.403) and WCSA change (−0.433) but not DBP change with the WCSA change (0.357, *p* > 0.005) (see Table 3).

The SBP change was correlated with WLR before (0.405) and after exercise (−0.471) but with a low R^2^ adjacent value (respectively, R^2^ = 0.190 and R^2^ = 0.291), whereas DBP change was associated only with WCSA change (0.441, R^2^ = 0.194). These associations were confirmed in the multivariate model, except for DBP as a predictor of WCSA change, achieving a coefficient R^2^ of 0.289.

There were significant differences in BMI and SBP before and after exercise between men and women (for BMI 24.2 vs. 22.2; *p* = 0.031; for SBP before 137.7 vs. 124.4; *p* = 0.009; for SBP after 168.5 vs. 152.8 (*p* = 0.009), for men vs. women, respectively). All other analysed parameters did not differ depending on sex.

Figure 2 and Figure 3 show the rtx1^TM^ AO retinal camera report with triplicate vessel estimations of a selected supratemporal artery before and after cessation of physical exertion in a patient. The artery change was calculated in the exact location of selected regions of interest (yellow squares).

## 4. Discussion

The eye is the only organ in which we can non-invasively visualise the smallest arterioles and analyse their response to different factors. The autoregulation system allows stable microcirculation in three vital organs: the heart, brain, and eye [12]. Retinal vessels are under autoregulation mechanisms to maintain a constant retinal blood flow [26]. The physiological regulation of the retinal microcirculation during exercise is complex. The available data show that physical exercise positively affects the retinal arterioles and lowers the risk of cardiovascular incidents [2,10,11,12,13,14,15,16,17,18,19,20,21,22,23]. The submaximal dynamic effort is associated with increased heart rate, systolic blood pressure, and cardiac blood flow, which was also observed in our study. Strategic redistribution of the blood flow to skeletal muscles and the heart with vasodilation of their vessels occurs during endurance exercises, while vasoconstriction is observed in skin and splanchnic tissues [32].

How exhaustive dynamic exercises influence the retinal arterioles is controversial. We hypothesised that retinal arterial vasoconstriction is seen during and immediately after vigorous physical activity. It was confirmed in our study, where vessel diameter and wall thickness decreased slightly but with statistical significance. Dumskyj MJ et al. have used Laser Doppler velocimetry to study retinal blood flow in response to an acute elevation of systemic blood pressure induced by isometric exercise (26). Autoregulation was demonstrated by a rise in flow in response to increased perfusion pressure using data derived from arteries and veins. Arteries were constricted by 3.4% (*p* = 0.002), and veins were dilated by 1.6% (*p* = 0.02) [26]. A protocol similar to ours was used by Ikemura T et al. in 12 healthy male subjects, with cycle ergometer exercise at 75% of maximal heart rate until exhaustion [32]. Blood flows in the retinal and choroidal vasculature were measured every 3 min during the exercises [32]. The study showed that, due to the hypocapnia at exhaustion, the retinal blood flow was significantly decreased [32]. Hayashi et al. also analysed retinal blood flow stability during cycling dynamic exertion in young, healthy participants [27]. They found that retinal arterioles’ blood flow and vascular conductance remained stable, while ocular blood perfusion increased in the retina and choroid during physical exercise [27]. It was the first study demonstrating the autoregulation of blood flow in retinal arterioles despite increased blood perfusion in the retina and choroid during dynamic exercise [27].

Iester M et al. also confirmed that autoregulation maintains stable blood after intense exercises in young participants [29]. Our study did not measure the retinal blood flow but the state of the same arteriole before and immediately after exercise. Only constriction of small arteries makes it possible to keep stable blood flow with increasing heart rate and systolic blood pressure. Harris A. et al. presented results similar to ours [28]. After exhaustion exercises, they found a significantly narrowed superior temporal artery and vein and unchanged calculated retinal blood flow [28]. It showed that vasoconstriction to normalise flow in the capillary bed in response to increased blood and perfusion pressure occurred as the retinal hemodynamic response [28]. They analysed retinal vessel parameters with an invasive examination, fluorescein angiography, which is more challenging to perform in every patient. In another study, Ikemura T. et al. showed that the retinal blood flow reduction induced by exhaustive exercise could be recovered early by rehydration [33]. In both groups, one rested without drinking (control, CON), and the second rested while drinking electrolyte-containing water (rehydrate condition, REH), blood flow decreased immediately after exercise from the resting baseline. The blood flow recovered to baseline at 15 min of recovery in REH but remained reduced in CON [33].

Retinal microcirculation analysis may be applied to monitor treatment strategies such as exercise interventions in a personalised medicine approach. High-resolution, high-contrast images of the retina obtained with the rtx1^TM^ AO retinal camera allowed for a detailed assessment of vessel dynamics. The AO technique made visualising vessel walls and the lumen possible [6]. This method lets us differentiate between an arteriole’s functional vasoconstriction and a vessel’s structural remodelling. The possibility of exact measurements of all vessel elements and control of their changes in time makes AO technology the ideal candidate as the vascular biomarker of cardiovascular risk prognosis. Among all vessel parameters, the WLR measured by SLDF or AO is now a promising indicator of retinal arteriolar morphological remodelling [6,10,11]. De Ciuceis et al. observed no correlation between WLR and systolic and diastolic blood pressure in 237 subjects, both normotensive and with hypertension [11]. A significant correlation between WLR and systolic blood pressure was observed only in untreated hypertensive patients. Our study found a significant correlation between changes in SBP and WLR after exercise. During the physical effort, the SBP increased significantly, which may be compared to untreated hypertensive patients in De Ciuceis’ study. In the long-term analysis of the risk of cardiovascular events, only the WLR of retinal arterioles has been the essential prognostic predictor of the occurrence of events [11]. High SBP is the expression of cardiac output and is associated with eutrophic arteriolar wall remodelling with no significant differences in WCSA, characteristic of hypertensive retinopathy [34].

During high-intensity exercise, the myogenic autoregulation resulting in retinal arteriole vasoconstriction is aggravated by hypocapnia induced by low arterial CO_2_ pressure [32]. It is the physiological mechanism of autoregulation seen in young people, declining with age, which was confirmed by Nussbaumer M et al. [24] and in our results. We analysed retinal microcirculation response only in the group of young volunteers. Decreased myogenic response and vascular reactivity loss in response to high exercise intensities in older people have been potential risk factors for retinal and cerebral haemorrhages in response to intraluminal pressure increase occurring with ageing [24]. It is a significant finding confirming the role of retinal microcirculation examination in assessing the risk of cardiovascular events.

The positive correlation between the mean BMI and the mean WLR measured before exercise was shown in our study. Jeppesen et al. showed a significant decrease in the retinal arteriolar diameter with increasing weight in healthy volunteers up to 40 but not in older groups [31]. rtx1^TM^ AO retinal camera was used to analyse the relation between BMI and retinal photoreceptor morphology and microvascular changes in healthy women [35]. Statistically significant differences in the retinal arteriole wall thickness in women with greater BMI were observed [35]. The mean WLR and WCSA parameters were also significantly higher in participants with overweight and obesity [35]. The higher the BMI, the greater the risk of microvascular dysfunction even before the onset of hypertension. The results are consistent with those obtained in our study, where we included healthy, young participants with BMI in normal ranges. Hughes AD et al. described an inverse relationship between arteriolar diameter (CRAE) and age and SBP in healthy subjects [36]. They also showed that elevated blood pressure and higher BMI were positively correlated with arteriolar narrowing [36]. The analysis of retinal vessels was carried out based on semiautomated SVA with CRAE, CRVE, and AVR equivalents [36]. Measuring all vessel parameters was impossible, as we used AO technology.

## 5. Limitations

The study was carried out in young, healthy subjects to show the normal physiological response to increased blood pressure and high heart rate on retinal microcirculation. It might be interesting to know if there will be the same findings in different age groups and also in patients with cardiovascular diseases like hypertension, diabetes, or dyslipidemia. We did not analyse blood flow in retinal vessels. However, the vital part of this study is the use of adaptive optics and measurements of the same artery. A joint study involving cardiologists and ophthalmologists would be interesting for exploring the retinal autoregulation mechanisms in different vascular disorders.

## 6. Conclusions

Our study shows that measuring retinal microcirculation changes in response to endurance exercises using the AO retinal camera is a promising technique for investigating the physiological effects of exercise on the eye. The study confirms that intensive dynamic physical exertion increases systolic blood pressure and heart rate and causes the vasoconstriction of small retinal arterioles, representing the physiological mechanism of myogenic autoregulation in the retina. Making measurements with AO rtx1^TM^ in the exact locations in subsequent examinations allows for tracking changes throughout the observation. AO rtx1^TM^ imaging allows a better understanding of how physical activity affects the eyes.

## Figures and Tables

**Figure 1 diagnostics-14-00710-f001:**
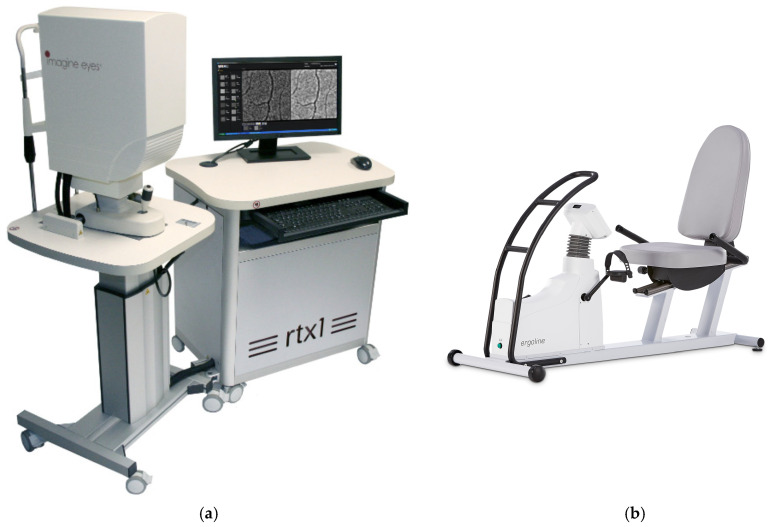
(**a**) The first panel presents the rtx1^TM^ AO—retinal camera; (**b**) the second panel shows the ergometer ergoline 600.

**Figure 2 diagnostics-14-00710-f002:**
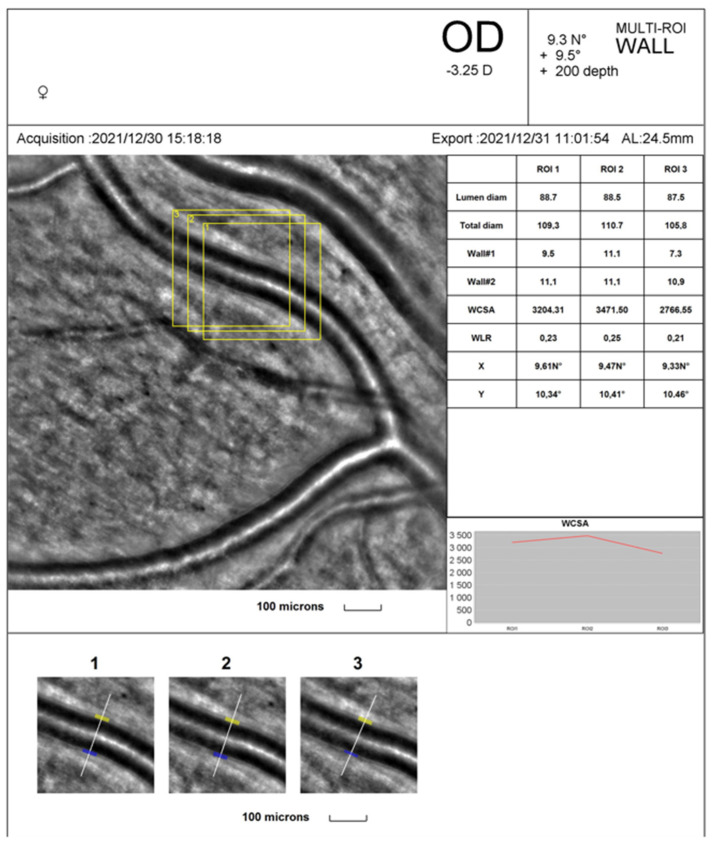
The rtx1^TM^ AO retinal camera analysis of a selected supratemporal artery before physical exertion.

**Figure 3 diagnostics-14-00710-f003:**
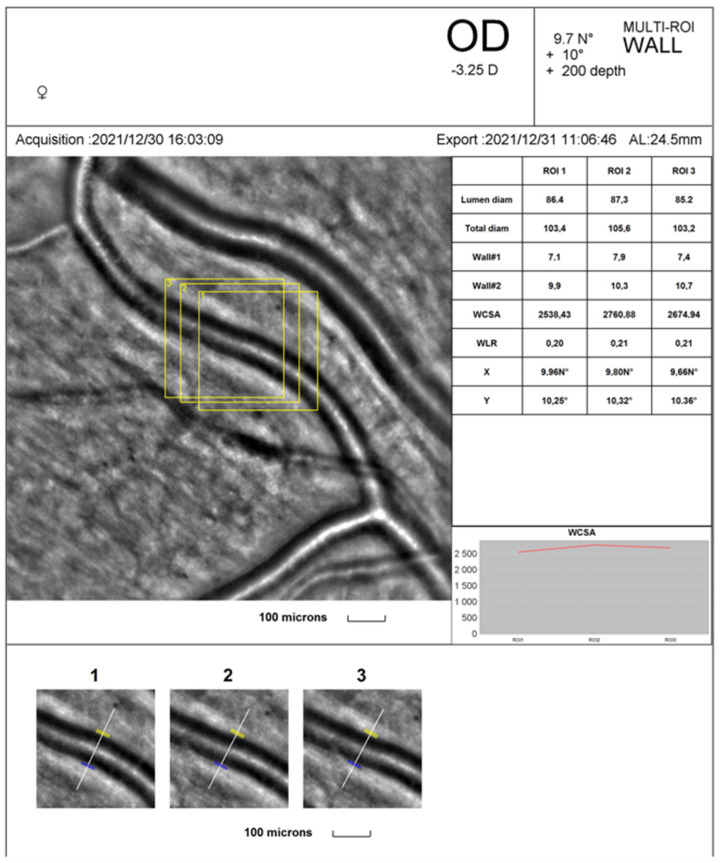
The rtx1^TM^ AO retinal camera analysis of a selected supratemporal artery after cessation of physical exertion.

**Table 1 diagnostics-14-00710-t001:** Cardiovascular parameters at rest and after physical exercise.

Parameters	Before Exercise (m ± SD)	After Exercise (m ± SD)	*p*-Value
SBP (mm Hg)	130.2 ± 13.2	159.7 ± 15.6	<0.001
DBP (mm Hg)	81.2 ± 6.3	77.1 ± 8.2	0.022
HR (bpm)	80.8 ± 16.1	175.0 ± 6.2	<0.001

m—mean; SD—standard deviation; HR—heart rate; SBP—systolic blood pressure; DBP—diastolic blood pressure.

**Table 2 diagnostics-14-00710-t002:** Characteristics of AO supratemporal retinal artery parameters at rest and after physical exercise.

Parameters	Before Exercise (m ± SD)	After Exercise (m ± SD)	*p*-Value
LD (μm)	96.0 ± 6.8	94.9 ± 6.7	0.258
VD (μm)	118.5 ± 8.3	115.9 ± 8.3	0.047
WT1(μm)	11.0 ± 1.5	10.4 ± 1.5	0.107
WT2(μm)	11.5 ± 1.0	10.7 ± 1.3	0.017
WLR	0.234 ± 0.02	0.222 ± 0.01	0.046
WCSA (μm^2^)	3802.8 ± 577.6	3512.3 ± 535.3	0.016

m—mean; SD—standard deviation; WT—wall thickness (1 and 2); WLR—wall-to-lumen ratio; LD—lumen diameter; VD—vessel diameter; WCSA—cross-sectional area of the vascular walls.

**Table 3 diagnostics-14-00710-t003:** Univariate and multivariate regression analysis of vessel parameters for independent parameters of BMI, SBP, and DBP changes.

Parameters	Univariate Analysis						General Linear Multivariate Analysis				
	BMI	R^2^	∆SBP	R^2^	∆DBP	R^2^	BMI	∆SBP	∆DBP	R^2^	R^2^_adj_
WLR_before	0.423 *	0.183	−0.405 *	0.190	0.062	<0.001	0.403 *	−0.459 *	0.116	0.380	0.291
∆WCSA (μm^2^)	0.366	0.237	0.156	0.025	0.441 *	0.194	0.433 *	0.063	0.357	0.378	0.289
WLR_after	0.039	<0.001	−0.471 *	0.291	−0.146	0.148	0.011	−0.543 *	0.015	0.291	0.190

* *p*-value < 0.05; m—mean; ∆—the value change before and after exercise; WLR—wall-to-lumen ratio; WCSA—cross-sectional area of the vascular walls; R^2^—coefficient of determination in regressions; R^2^_adj_—adjusted coefficient of determination in the multivariate regression model.

## Data Availability

The data presented in this study are available on request from the corresponding author.

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
