# Peer review of "Retinal Microcirculation Measurements in Response to Endurance Exercises Analysed by Adaptive Optics Retinal Camera"

_diagnostics, 2024, doi:10.3390/diagnostics14070710_

Round 1
Reviewer 1 Report
Comments and Suggestions for Authors
Manuscript id: diagnostics-2891366
Retinal Microcirculation Measurements in Response to Endurance Exercises Analysed by Adaptive Optics Retinal Camera
The aim of this study is to investigate the effects of intensive physical submaximal physical exercise on retinal microvascular regulation in healthy volunteers by adaptive optics in healthy volunteers. For this purpose, they assessed morphological parameters of retinal arterioles before and immediately after exercise and they reported an increase in SBP and HR and a reduction in DBP. They found a reduction of vessel diameter, wall thickness, WLR and WCSA with a slight change in lumen diameter.
The topic is interesting to the field, methods and language are fine. However, there are some points which need to be addressed or clarified.
1) The results about retinal arteriole structure as reported in the abstract are not clear.
2) Which were the habits in physical activity of the subjects. Do they were sedentary or physically active?
3) The Authors included both men and women. Do they observe a difference between blood pressure response or structural parameters of retinal arterioles between male and female?
4) Usually there is a reduction in vessel diameter and a parallel increase in wall thickness when vasoconstriction occurs and that results in reduction of WLR. In this study, the reduction of retinal WLR is due to a reduction of wall thickness with no change in lumen diameter. How do the Authors explain the reduction of wall thickness in presence of vasoconstriction?
Comments on the Quality of English Language
The results about retinal arteriole structure as reported in the abstract are not clear and they are diffult to understand.
Author Response
Dear Reviewer
There are responses to your comments and questions:
1. The results about retinal arteriole structure as reported in the abstract are not clear.
We added numeric results of the rtx1 analysis in the abstract ( lines 22-26). Due to the limited number of words in the abstract, we changed some other sentences.
Retinal microcirculation analysis showed no significant decrease in LD, Wall1 after exercise: from 96.0 ± 6.8 to 94.9 ± 6.7 (p = 0.258), from 11.0 ± 1.5 to 10.4 ± 1.5 (p = 0.107), respectively and significant reduction in VD from 118.5 ± 8.3 to 115.9 ± 8.3 (p = 0.047), Wall2 from 11.5 ± 1.0 to 10.7 ± 1.3 (p = 0.017), WLR from 0.234 ± 0.02 to 0.222 ± 0.010 (p = 0.046), WCSA from 3802.8 ± 577.6 to 3512.3 ± 535.3 (p = 0.016).2. Which were the habits in physical activity of the subjects. Do they were sedentary or physically active?
We added to the text the following new lines: 154-156.
Study participants were healthy volunteers who led an active lifestyle. None of them practised sports professionally in the past or currently. 3. The Authors included both men and women. Do they observe a difference between blood pressure response or structural parameters of retinal arterioles between male and female? We added the following new lines to the text: 250-262 and new Table 3.There were significant differences in BMI and SBP before and after exercise between men and women (for BMI 24.2 vs 22.2; p = 0.031; for SBP before 137.7 vs 124.4; p = 0.009; for SBP after 168.5 vs 152.8 (p = 0.009) respectively men vs women. All other analysed parameters did not differ depending on sex.
Table 3. Univariate and multivariate regression analysis of vessel parameters for independent parameters of BMI, SBP and DBP changes
|
Parameters |
Univariate analysis |
General linear multivariate analysis |
||||||||||
|
BMI |
R2 |
∆SBP |
R2 |
∆DBP |
R2 |
BMI |
∆SBP |
∆DBP |
R2 |
R2 adj |
||
|
WLR_before |
0.423* |
0.183 |
-0.405* |
0.190 |
0.062 |
< 0.001 |
0.403* |
-0.459* |
0.116 |
0.380 |
0.291 |
|
|
∆WCSA (μm 2) |
0.366 |
0.237 |
0.156 |
0.025 |
0.441* |
0.194 |
0.433* |
0.063 |
0.357 |
0.378 |
0.289 |
|
|
WLR_after |
0.039 |
< 0.001 |
-0.471* |
0.291 |
-0.146 |
0.148 |
0.011 |
-0.543* |
0.015 |
0.291 |
0.190 |
|
4. Usually, there is a reduction in vessel diameter and a parallel increase in wall thickness when vasoconstriction occurs, and that results in a reduction of WLR. In this study, the reduction of retinal WLR is due to a reduction of wall thickness with no change in lumen diameter. How do the Authors explain the reduction of wall thickness in presence of vasoconstriction?
We added these sentences to the text (lines 230-235)
The contraction of smooth muscles in the retinal arteriole’s wall, which leads to a reduction of its thickness, is involved in the myogenic mechanism of retinal autoregulation in response to changes in pressure. There was a significant decrease in vessel diameter but not a significant lowering of lumen diameter. Thickening of arterioles walls is found in vessel remodelling due to vascular and metabolic diseases but not in physiological mechanisms. 5. The results about retinal arteriole structure as reported in the abstract are not clear and they are diffcult to understand. The abstract has been improved, as well as some other part as above noted.Reviewer 2 Report
Comments and Suggestions for Authors
In the manuscript, the authors used AO rtx1camera to measure the retinal microcirculation changes for investigating the physiological effects of exercise on the eye and allows for tracking changes throughout the observation. The healthy volunteers were performed and the idea is innovative and practical. The discussion part looks great! The reviewer thinks that would be attractive to the audience as a good tool to measure retina during excersie. However, there are some minor revisons that authors need to be addressed before publish.
1. Figures in the manuscript are not sufficient. Authors might use Table 1 and Table 2 to draw a column chart or bar chart with a P-value for better demonstration of the results.
2. The artery change in Fig.2 is not obvious. Authors might zoom in the subfigure 1-3 and label the change more carefully.
3. Please check the journal's template. Convert it to the current journal's template.
Comments on the Quality of English Languagemoderate revision.
Author Response
Dear Reviewer,
There are responses to your comments and questions.
1. Figures in the manuscript are not sufficient. Authors might use Table 1 and Table 2 to draw a column chart or bar chart with a P-value for better demonstration of the results.
The authors suggest that tables 1 and 2 may be left in these forms as they consist of more data in a more condensed form.
2. The artery change in Fig.2 is not obvious. Authors might zoom in the subfigure 1-3 and label the change more carefully.
We have added more detailed descriptions of zoomed figures 2 and 3 in the text (lines 264-267) and added new captions to these figures.
Figures 2 and 3 show the rtx1 TM AO-retinal camera report with triplicate vessel estimations of a selected supratemporal artery before and after cessation of physical exertion in a patient. The artery change was calculated in the exact location of selected regions of interest (yellow squares).Figure 2. The rtx1 TM AO-retinal camera analysis of a selected supratemporal artery before physical exertion.
Figure 3. The rtx1 TM AO-retinal camera analysis of a selected supratemporal artery after cessation of physical exertion.
The authors have made figures 2 and 3 bigger, leaving the original form of a rtx1 report with the anonymized name of a patient.
3. Please check the journal's template. Convert it to the current journal's template.The manuscript has been successfully converted using the MDPI template into proper form. We have also checked and corrected other minor problems with spaces, dot and comma questions, etc.
We have improved the English language in the manuscript.